# A Novel Mechanical Fault Diagnosis Based on Transfer Learning with Probability Confidence Convolutional Neural Network Model

**Hsiao-Mei Lin [1,*], Ching-Yuan Lin [1], Chun-Hung Wang [2] and Ming-Jong Tsai [2,*]**

1   Department of Architecture, National Taiwan University of Science and Technology, Taipei 106335, Taiwan
2   Graduate Institute of Automation and Control, National Taiwan University of Science and Technology, Taipei 106335, Taiwan
*   Correspondence: d10413005@mail.ntust.edu.tw (H.-M.L.); mjtsai@mail.ntust.edu.tw (M.-J.T.)

**Abstract:** For fault diagnosis, convolutional neural networks (CNN) have been performing as a data-driven method to identify mechanical fault features in forms of vibration signals. However, because of CNN's ineffective and inaccurate identification of unknown fault categories, we propose a model based on transfer learning with probability confidence CNN (TPCCNN) to model the fault features of rotating machinery for fault diagnosis. TPCCNN includes three major modules: (1) feature engineering to perform a series of data pre-processing and feature extraction; (2) transferring learning features of heterogeneous datasets for different datasets to have better generality in model training and reduce the time for modeling and parameter tuning; and (3) building a PCCNN model to classify known and unknown fault categories. In addition to solving the problem of an imbalanced sample size, TPCCNN self-learns and retrains by iterating with unknown classes to the original model. This model is verified with the use of the open-source datasets CWRU and Ottawa. The experimental results showing the feature transfer of heterogeneous datasets are of average accuracy rates of 99.2% and 93.8% respectively for known and unknown categories, and TPCCNN is then proven effectively in training heterogeneous datasets. Likewise, similar feature sets can also be applied to reduce the training of predicting models by 34% and 68% of the time.

**Keywords:** fault diagnosis; probability confidence; feature engineering; transfer learning; deep learning; convolutional neural networks

## 1. Introduction

Rotating machinery plays an essential role in many industries. Bearings are among the most common components for rotating machinery. The stability of automated machinery and equipment is one of the crucial factors affecting factory production. For example, mechanical vibration can cause bearing damage, spindle eccentricity of rotating machinery, and the damage and failure of equipment. Thus, exploring the precise approaches for fault diagnosis is of great value because unpredictable faults of machinery can lead to severe damage and losses in production. In fault diagnosis, most defects are resulted from equipment vibration. The vibration of bearings is often used as a means of fault diagnosis and prediction of the equipment's remaining life to improve the equipment's stability and reduce economic losses [1].

In smart manufacturing, the application of intelligent processing equipment is a trend. This can be seen in the fact that more and more manufacturers utilize electromechanical integration, online monitoring, and value-added software technologies to improve the performance of machinery. Hence, the research and application of fault diagnosis are of interdisciplinary work [2]. X. Zhou et al. [3] proposed 1D convolutional neural network fusing frequency domain feature matching algorithm (FDFM) to learn the crucial features directly from the frequency domain, and perform fault identification under limited samples

and a noisy interference environment. S. Xiong et al. [4] proposed an end-to-end fault diagnosis of rolling bearings by wavelet packet transform (WPT) and CNN methods. Generally, there are generally two approaches to fault diagnosis—data-driven methods and fault-based modeling. The traditional data-driven models of fault diagnosis rely on experts' domain knowledge about the corresponding mechanical parts. Besides, the work of exploring various machines and fault classification is labor-intensive and time-consuming [5]. The complexity of electromechanical systems building rotating machinery makes good modeling for fault diagnosis a much more challenging work. Thus, a deep learning-based data-driven model of fault diagnosis has been developed as it captures signal features automatically and rely on no professional knowledge of humans. Standard models of deep learning applied to fault diagnosis models include convolutional neural networks (CNNs), autoencoders, recurrent neural networks (RNNs), and generative adversarial networks (GANs) [6]. C. Kuo et al. [7] proposed a practical rotor failure diagnostic method with fuzzy theory and a genetic algorithm for evaluating operational status of motors. X. Wang et al. [8] proposed a prediction method—bearing remaining useful life (RUL)—that took both time-domain features and time-frequency features into account on the basis of parallel deep residual convolution neural network (P-ResNet) to raise the prediction accuracy. J. Zhou et al. [9] proposed a residual network, which combined transfer learning (ResNet-TL) based diagnosis methods of rolling bearings, and was able to preprocess one-dimensional data of vibration signals into image data for the application of transfer learning afterwards to pre-train and re-train the ResNet34 network. Z. Xu et al. [10] proposed a text-driven fault diagnosis model based on Word2vec, CNN, and CSM. To extract the text extraction using Word2vec and build the prior-knowledge CNN classifier with Cloud Similarity Measurement (CSM) improved the accuracy of aircraft fault diagnosis. J. Chuya-Sumba et al. [11] proposed a 1D CNN model that works on raw signals without any need of prerequisite analysis. G. Nassajian and S. Balochian [12] proposed a multi-model estimation and fault detection method using RBF neural network for a nonlinear system of unknown time continuous fractional order.

Testing with different operating conditions of equipment such as speed, load, environmental noise, and fault location, can result in uneven data distribution and unbalanced sampling [13]. The method of generative adversarial network (GAN) generates data deriving from learning different failure characteristics to expand the training data and solve the data imbalance problem [14]. However, most diagnostic models proposed so far are based on supervised learning that identifies labels [15], and identifying different types of fault data is more challenging. In the industry, it is a hard task to label fault data as it is regarded as an "unknown category," so simulation is usually applied to the faults of this type. A PCCNN algorithm [16] is used in the computation of probabilistic confidence levels to distinguish between "known classes" and "unknown classes" of failure classes. However, the current research is still applied as a primary source in the open-source simulation dataset for the low simulation efficacy in different cases.

Most previous studies based on the method of data-driven intelligent fault diagnosis (DIFD) focused on the improvement of the generalization performance and fault diagnosis with several reconfigurations. Zheng et al. proposed [17] domain adaptation from transfer learning and other techniques to achieve cross-domain fault diagnosis. Yan et al. [18] provided an overview of knowledge transfer for rotary machinery fault diagnosis (RMFD) by applying different transfer learning techniques in four categories: transfer between multiple fault classes, transfer between numerous locations, transfer between working conditions, and transfer between various machines. Different machines have different failure classes and data characteristics. Sun et al. [19] proposed transfer learning based on stacked autoencoders (SAEs) algorithms combined with classification and domain-blending to improve the accuracy of diagnostic models and the versatility of fault diagnosis data for different machines. In preceding research, models of fault diagnosis were established with the combination of transfer learning and deep neural networks. Based on the results, the research carries out the solution to the imbalance of sample fault data. Fan Yang et al. [20]

proposed two transfer strategies to analyze the probable scenarios in practical cases and suggested transfer strategies applicable in each case.

In previous works, the fault-based modeling, and data-driven methods for known fault diagnosis has been performed, while the authors of this paper made a preliminary survey of fault detection in data-driven with unknown class and transfer-learning in similar datasets. This paper proposed TPCCNN (Transfer PCCNN) focusing on monitoring vibration frequencies which can be featured by FFT and trained and transferred in PCCNN model for further fault diagnosis for the first time.

The rest of this paper is organized as follows:

The Section 2 presents the principle of TPCCNN, the introduction of PCCNN, and the method of TPCCNN-based fault diagnosis including feature extraction, pre-trained model, and fine-tuning. This is followed by a presentation about the experimental setting, datasets, processes, and results of TPCCNN. The Section 4 gives the experimental results to demonstrate the efficiency of the proposed method. Finally, the paper gives conclusion and future work.

## 2. Materials and Methods

### 2.1. Principle of Transfer Learning in TPCCNN

Several features, vibration frequencies, abnormal noise, etc., can be derived from the fault of a rotary machinery. TPCCNN (Transfer PCCNN) proposed in this paper focuses on monitoring vibration frequencies which can be featured by FFT, trained, and transferred in PCCNN model for further fault diagnosis. Due to different operational conditions and environments, subtle bias happens between machines with the same model and among different machines which affect the accuracy of model evaluations. Therefore, a method of TPCCNN combined with transfer learning and the PCCNN [16] models is developed for fault diagnosis of rotary machinery. The architecture diagram of TPCCNN is shown in Figure 1. The TPCCNN model derives from PCCNN that consists of four convolutional layers, four pooling layers, and three fully connected layers. We fine-tuned the preceding model. While adjusting, we used a learning rate that was equal to or less than the one used in the initial training model. We rarely did adjustment on the defined weights, for we had been highly confident in the pre-trained network.

The transfer learning model in this study adopts parameter transfer, which fixes the features of the lower convolutional layers, pooling layers, and batch normalization before retraining the weights and parameters of the higher fully connected layers, as shown in Table 1.

**Table 1.** Details of Structural Parameter of the TPCCNN Model.

| Layer | TPCCNN Parameter | | | |
| --- | --- | --- | --- | --- |
| | Parameter Size | Activation Function | Batch Normalization (BN) | Freeze/Fine-Tune |
| Input | / | / | / | / |
| Convl-1 | $32 \times 64 \times 1 \times 1 \times 4$ | ReLU | Yes | Freeze |
| Pooling-2 | $2 \times 1 \times 2$ | / | No | Freeze |
| Convl-3 | $64 \times 3 \times 1 \times 32 \times 1$ | ReLU | Yes | Freeze |
| Pooling-4 | $2 \times 1 \times 2$ | / | No | Freeze |
| Convl-5 | $96 \times 3 \times 1 \times 61 \times 1$ | ReLU | Yes | Freeze |
| Pooling-6 | $2 \times 1 \times 2$ | / | No | Freeze |
| Convl-7 | $128 \times 3 \times 1 \times 96 \times 1$ | ReLU | Yes | Freeze |
| Pooling-8 | $2 \times 1 \times 2$ | / | No | Freeze |
| FullContd-9 | / | / | No | Fine-tune |
| FullContd-10 | $M \times 512$ | ReLU | Yes | Fine-tune |
| FullContd-11 | $512 \times N$ | / | Yes | Fine-tune |
| SoftMaxPlus-12 | N | / | No | Fine-tune |
| Output | / | / | / | / |

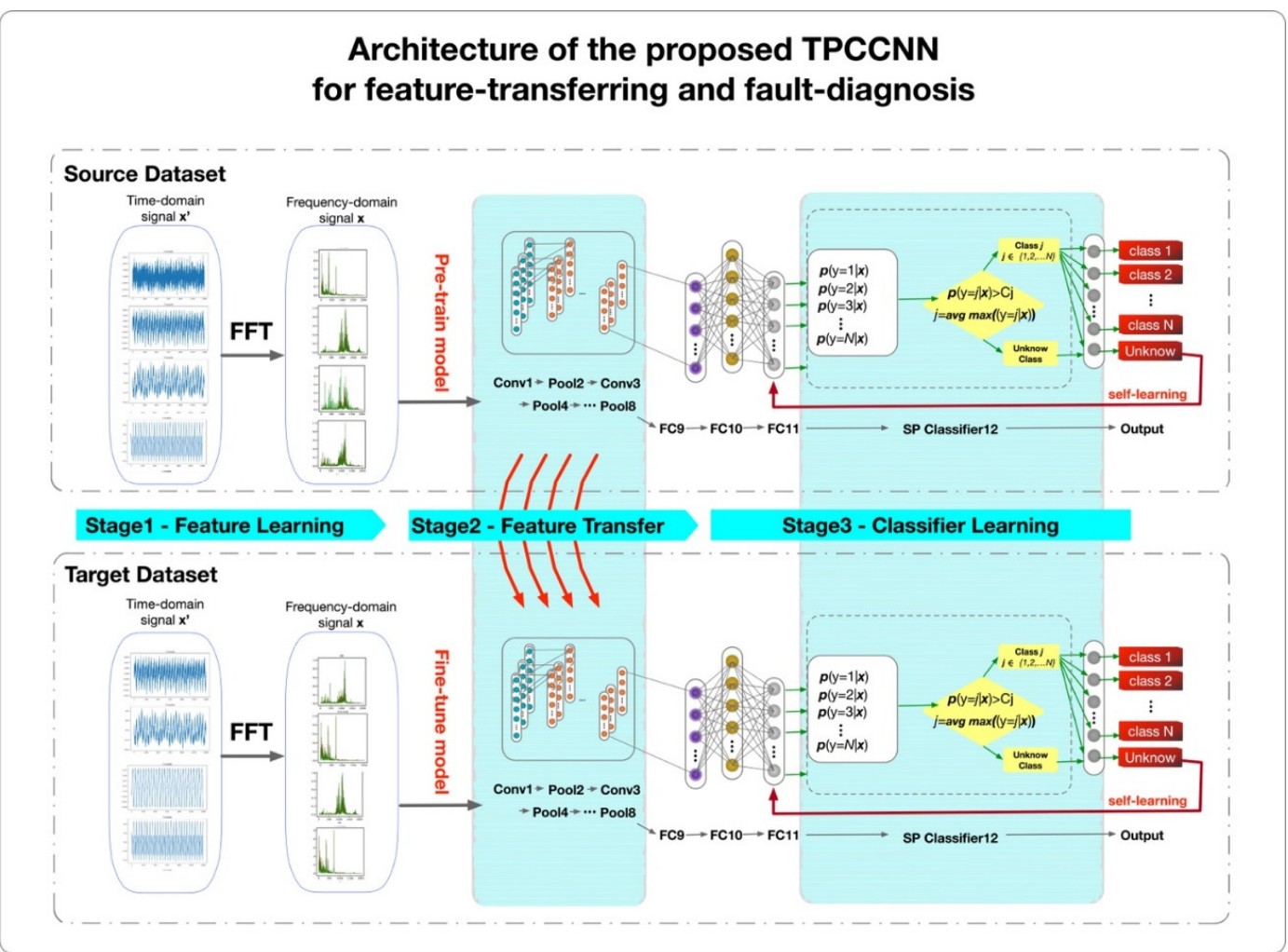

**Figure 1.** Architecture of the proposed TPCCNN framework.

*2.2. PCCNN*

A PCCNN algorithms [16], in which PC stands for Probability Confidence, are employed in CNN (Convolutional Neural Network) model to improve the accuracy. The architecture is shown in the right side of Figure 1. PCCNN is used in the computation of probabilistic confidence levels to distinguish between "known classes" and "unknown classes" of failure classes. First, being initialized with a set of labeled training data, the system calculates the confidence interval and probability of each known class to evaluate the reliability probability of the statistical inference. Significance is referred to as the probability in which the estimated parameter falls within a specific range when making statistical inferences. Second, PCCNN has a self-learning ability. The threshold values of each category comprise recorded in a vector $C$ and is defined as the probability threshold value within the normal range. The lower limit is set at the threshold value $C$ to distinguish the category of known faults from that of unknown faults. Therefore, given that a value exceeds 1.5 times the range of the 1st and 3rd quartile range, i.e., $1.5 \times IQR$, it is classified as an outlier and placed in the unknown category. The vector $C \in \mathbb{R}^N$, where N is the number of know classes, representation of probability confidence is shown in Equation (1).

$$C_j = Q1 - 1.5 \times (Q3 - Q1) \tag{1}$$

Not only the data but also the detection and recognition models need to be kept up-to-date to improve the adaptability of the diagnostic model and to reduce diagnostic errors.

Given the index number of unknown categories reaches a specific value, the unknown type is identified, and the index shifts by one to the $N+1$th category. Substituting the known categories into the model, training, and adjusting to identify the $N+1$th new category promote model optimization and adaptation.

### 2.3. TPCCNN-Based Fault Diagnosis

This fault diagnostic model architecture includes data pre-processing, model pre-training, and model fine-tuning. Time series classification is an essential field in time series data mining. It has been widely used in different areas, such as medical science electrocardiogram for health diagnosis, identification of human activities, and computer science for speech recognition and machine fault detection. With the advent of deep learning, new methods were developed, especially convolutional neural network (CNN) models. Although it has drawn great interest in the past few decades, it is still challenging and inefficient due to the nature of its data: high dimensionality, large data volume, and constant updates. Lamyaa Sadouk et al. [21] have reviewed several techniques to deal with time series classification, which can be categorized as model-based, distance-based, and feature-based. Most deep learning architectures are unable to directly process the raw input data of vibration for final defect classification and prediction. Further, the TPCCNN-based model is unable to deal with the raw data. In order to enable end-to-end computation for deep learning architectures, data preprocessing techniques play a crucial role in intelligent fault diagnosis [22–24].

The original fault data of the source and target domain are obtained by the vibration sensor and presented as time-domain data. First, the fast Fourier transform (FFT) is applied to map the data into frequency domain, as shown in Equation (2). The frequency-domain signals have higher fault recognition accuracy than time-domain signals, and the frequency domain data is then normalized with the maximum and minimum values for normalization. That is, the data is scaled within the interval of $0 \leq X' \leq 1$. The calculation method is shown in Equation (3).

$$X_k = \sum_{n=0}^{N-1} X_n e^{-i2\pi k \frac{n}{N}}, k = 0, \, 1 \ldots, \, N-1. \tag{2}$$

$$X\prime = \frac{X - X_{min}}{X_{max} - X_{min}} \in [0, \, 1] \tag{3}$$

### 2.4. Pre-Trained Model

A pre-trained model is trained with a large dataset and typically applied to large-scale image classification. Given that the original dataset is sufficiently large and general, the spatial hierarchy of features learned by the pre-trained model is used for a general model. Its features are equally effective for different computer visions, even for identifying classes that are completely different from the original task. This approach is applied to time series problems with a similar effect. Compared with traditional machine learning methods, the key advantage of deep learning is that the learned features are transplanted to different problems, which makes the model reusable and effective in the cases of small samples [25].

### 2.5. Feature Extraction and Fine-Tuning

Two ways of pre-train model are used: feature extraction and fine-tuning. Feature extraction is a collection of representations learned by previous models to obtain useful features from new samples, which are then fed into new classifiers for training and inference. Fine-tuning is a variation of feature extraction. Taking a CNN model as an example, the lower layers in the model extract local and highly general feature maps while the higher layers extract more abstracted concepts. Therefore, when extracting features as knowledge transfer, the convolution-based part is usually used as the reusable part of the model. Feature exaction freezes all convolution-based layers, while fine-turning only freezes the part of mainly retraining high-level convolution-based and dense layers. Fine-tuning

requires training of top-level classifier for the new dataset in advance. If the classifier is not trained in advance, the error signal propagating through the network during training becomes too large and potentially corrupts what has been learned by previous fine-tuning layers. Since the backpropagation algorithm calculates the gradient of the loss function for each weight by the chain rule, the gradient of one layer is calculated one at a time and then iterates backward from the last layer [26].

## 3. Results

The proposed method is validated on two open-source datasets, which include the CWRU dataset and the Ottawa Mendeley dataset. The TPCCNN model is conducted by using python 3.7 which runs on a computer with CPU i9-11900@2.50 GHz, RAM 32 GB, and GTX 3070 GPU. The operating system is 64-bit Win11. Two datasets are used alternately as the source and target for model training and testing in the experiment. CWRU includes bearing failure data at different speeds and loads, while Ottawa contains bearing failure data at different rates. The information for the two publicly available datasets is detailed below as shown in Table 2.

**Table 2.** Comparison between CWRU with Ottawa.

| Items | | CWRU Dataset | Ottawa Dataset |
|---|---|---|---|
| **Health Condition** | Normal | v | v |
| | Inner Race Fault | v | v |
| | Outer Race Fault | v | v |
| | Ball Fault | v | - |
| **Sampling Frequency** | | 12,000 Hz | 200,000 Hz |
| **Dataset Size** | | 66.8 MB | 458 MB |
| **Data Length** | | 10 s | 10 s |
| **Shaft Speed** | | Avg. 1730 rpm (1720~1797 rpm) | (a) Increasing speed (b) Decreasing speed (c) Increasing then decreasing speed (d) Decrease then increasing speed |
| **Load** | | 0~3 hp | - |

### 3.1. Dataset

The CWRU Dataset is provided by society for machinery failure prevention technology (MFPT) [27]. This dataset contains ball bearings test data for both normal and faulty bearings. This dataset records the motor's actual test conditions and the bearing's failure status with different experimental data, as shown in Table 2. The Ottawa Mendeley Dataset is provided by the University of Ottawa in Canada [28]. This dataset contains vibration signals collected by bearings of different health conditions with varying speed conditions. There are 60 datasets in total. Each dataset has two experimental setups: bearing health and variable speed conditions.

### 3.2. Pre-Processing

Data preprocessing techniques play a key role in intelligent fault diagnosis to enable end-to-end computation for deep learning architectures. This research performs a series of data preprocessing and feature extraction, such as signal time-frequency domain conversion, noise reduction, and inductive bias. We use the FFT transformed vibration signal in frequency domain as the input of the one-dimensional convolutional neural network. The shift stride data augment trick increases the amount of processed vibration data. The relevant parameters are explained as follows:

1.  The unit length of original data X is 4096. That is, the time-domain data of 4096 points are sourced from the original acceleration vibration signal. FFT transformation then obtains a frequency domain of 4096 points.
2.  The unit length of frequency-domain data Y is 2048. There are 2048 points in the first half starting from low frequency. The points are selected from X to obtain new frequency domain data Y.
3.  The original acceleration vibration signal contains more than 4096 Z data points. We define a data interval of 512 points to group samples for processing. In other words, the sliding step size is 512 points as the data interval to separate the samples.

Multiple X data samples are obtained by intercepting sliding sampling, so a total of ((Z-4096)/512 + 1). For example, the data interval of configuration parameter of the CWRU is $64 \times 2$ and the Ottawa is $64 \times 12$. Since the total data volume of different data sets is inconsistent, this parameter is for different data sets.

Figure 2 shows the signal plots in frequency-domain for normal, inner race fault, outer race fault, and ball fault to demonstrate feature extraction. The feature can be identified. For example, the normal has only a low frequency under 1000 Hz. It is easy to understand and explain how the AI model classifies failure categories.

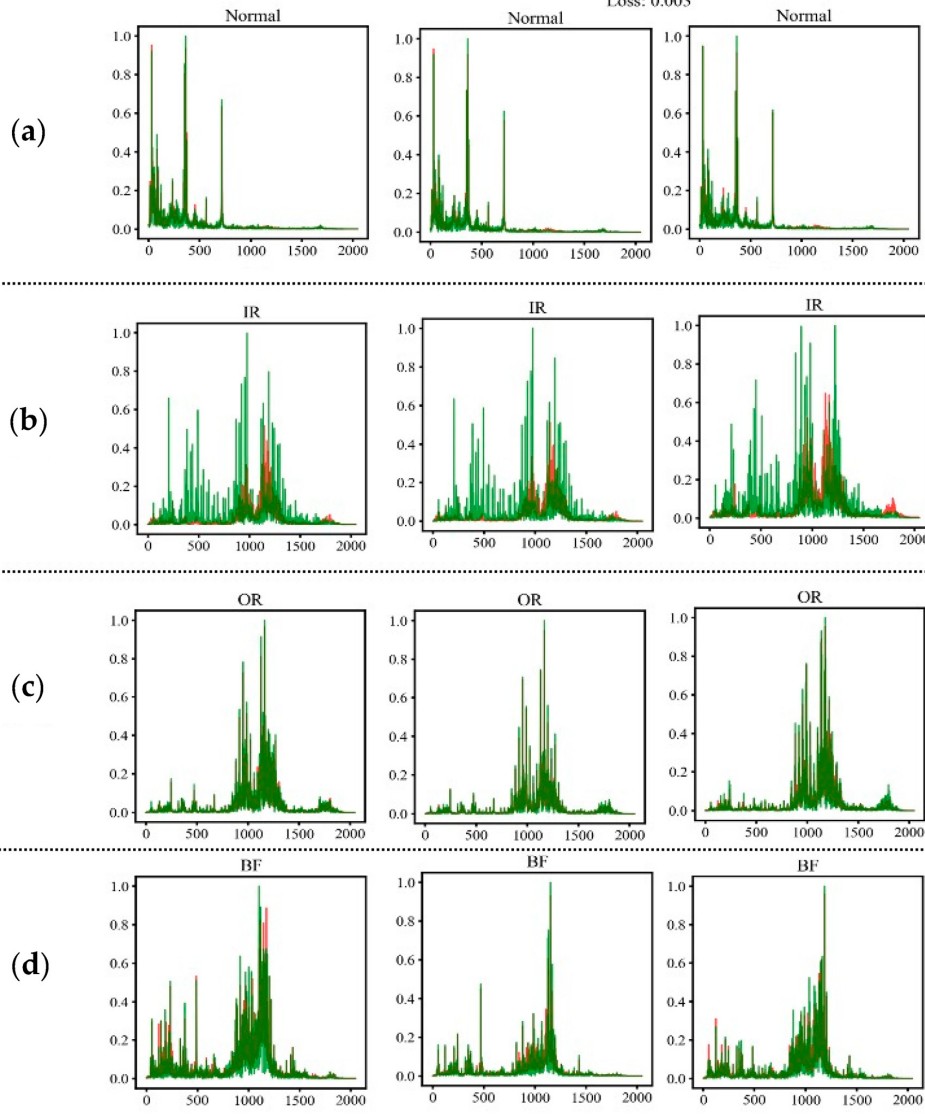

**Figure 2.** The feature extraction plots from CWRU dataset. (**a**) Normal. (**b**) Inner Race Fault. (**c**) Outer Race Fault. (**d**) Ball Fault.

### 3.3. Pre-Trained Model

Two models are trained and evaluated as the baseline for training and evaluation in the experiment. One uses the CWRU dataset in the constructed TPCCNN model for training while the other uses the Ottawa dataset for model training. In both pre-trained models, the configuration is set as follows. The learning rate is 0.001, the momentum is 0.9, the batch size is 32, the epoch is 30, and the RMSprop optimizer is selected for optimization.

### 3.4. Fine-Tune

In TPCCNN models, while extracting features as knowledge transfer, the convolution-based part is often used as a reusable part of the model because it has local and highly general feature maps. The experimental approach takes the architecture of a pre-trained model and then trains top layers while freezing others. The experiment contains three settings: (1) retraining only the output classifier, (2) retraining the densely connected classifier at the top layer, and (3) fine-tuning.

## 4. Discussion

The contribution of this paper to the feature reduction methods are aggregated into two categories: data-level and algorithm-level approaches. The data-level approach consists of encoding time series using FFT to clean and produce de-noised input signals which offer a more efficient CNN training. In the real world, if the spectrum with heavy noise, the FFT can efficiently clean the data and retrieve smooth results we expect.

In the algorithm level approach, one is the PCCNN algorithm which has a self-determined and self-learned ability to distinguish between unknown and known classes. The other is a transfer learning algorithm with adaptive convolutional layer filters and classifiers to analyze the input time series signals, including noise fluctuation.

According to the previous description, CWRU and Ottawa are used for model training and evaluation. The proportions of training and test sets for each dataset are 70% and 30%. Since the sampling lengths of each data set are different, the sliding sampling method is adopted. The sliding data interval of CWRU and Ottawa is set to 128 and 768, respectively. Experimental results are visualized using the confusion matrix and AUC/ROC curve.

Figures 3 and 4 show the confusion matrix and test results for transferring the extracted features from the source domain CWRU dataset to the target domain Ottawa dataset. The dataset of the target domain Ottawa has three health states: normal, faulty with an outer race defect (OR), and faulty with an inner race defect (IR). There are two sub-experiments to test the accuracy: (a) hidden the labels of inner race faults (IR) and (b) hidden the labels of outer race faults (OR). The dataset has 11,000 samples and is divided into the training set and test set according to the ratio of 7:3. Thus, there are 7700 samples in the training set and 3300 pieces of data in the test set.

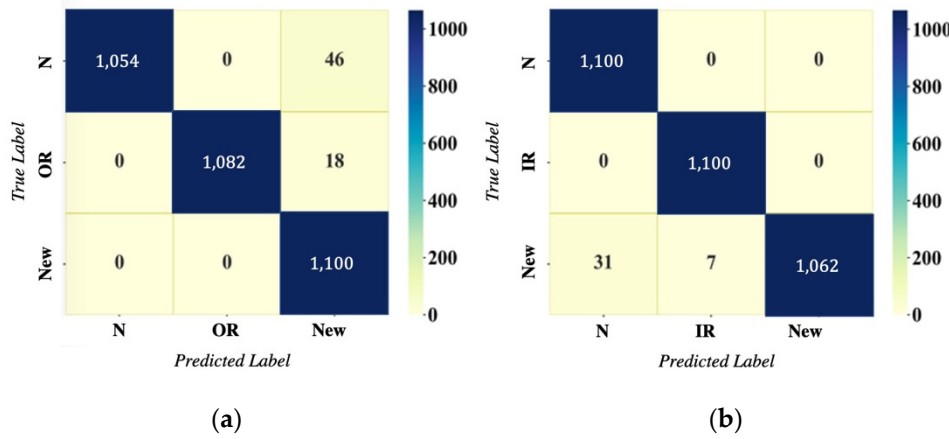

(**a**)        (**b**)

**Figure 3.** Confusion matrix of feature transferring from CWRU to Ottawa. (**a**) Hidden the labels of inner race faults (IR). (**b**) Hidden the labels of outer race faults (OR).

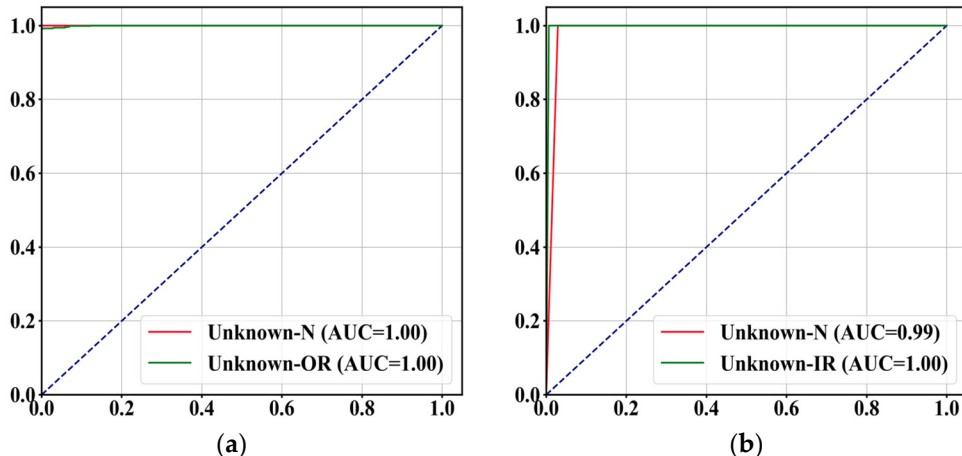

(**a**)  (**b**)

**Figure 4.** ROC/AUC of feature transferring from CWRU to Ottawa. (**a**) Hidden the labels of inner race faults (IR). (**b**) Hidden the labels of outer race faults (OR).

As the confusion matrix shows, when the new fault classes are different, the model still accurately recognizes the known and unknown classes. When OR is used as an unknown category, it has an accurate judgment of the data of the known category, and the AUC value is 1.00. When IR is used as the unknown category, the judgment of the known category is also accurate, and the AUC value is also 1.00. By taking different types of faults as unknown categories, the recognition degree of the model to different categories and the robustness of the model is stable.

Figures 5 and 6 show the confusion matrix and test results for transferring extracted features from the source domain Ottawa dataset to the target domain CWRU dataset. The target domain CWRU dataset has four health states: normal, outer ring bearing failure (OR), inner ring bearing failure (IR), and ball failure (BF). There are three sub-experiments to test the accuracy: (a) hidden the labels of ball faults (BRF), (b) hidden the labels of inner race faults (IR), and (c) hidden the labels of outer race faults (OR).

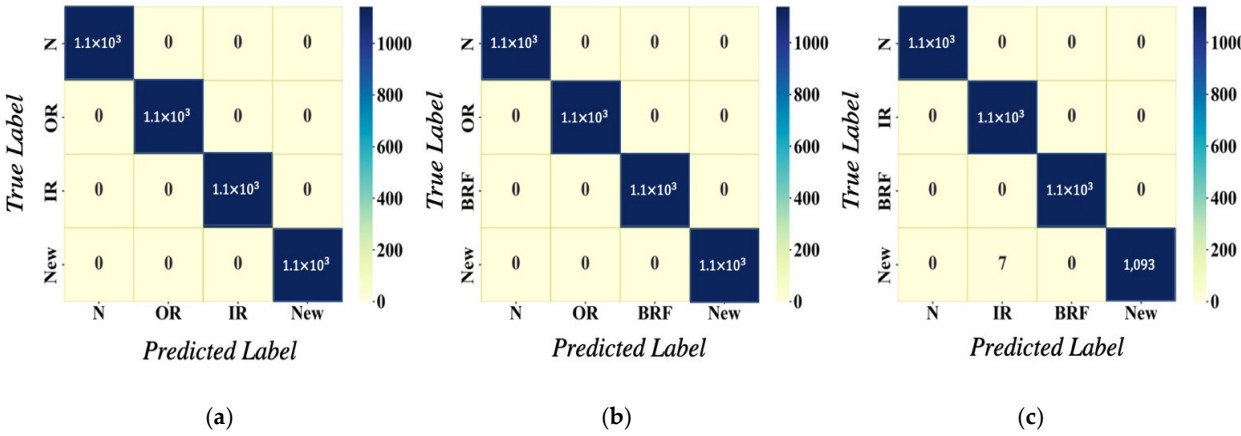

(**a**)  (**b**)  (**c**)

**Figure 5.** Confusion matrix of feature transferring from Ottawa to CWRU. (**a**) Hidden the labels of BRF faults. (**b**) Hidden the labels of IR faults. (**c**) Hidden the labels of OR faults.

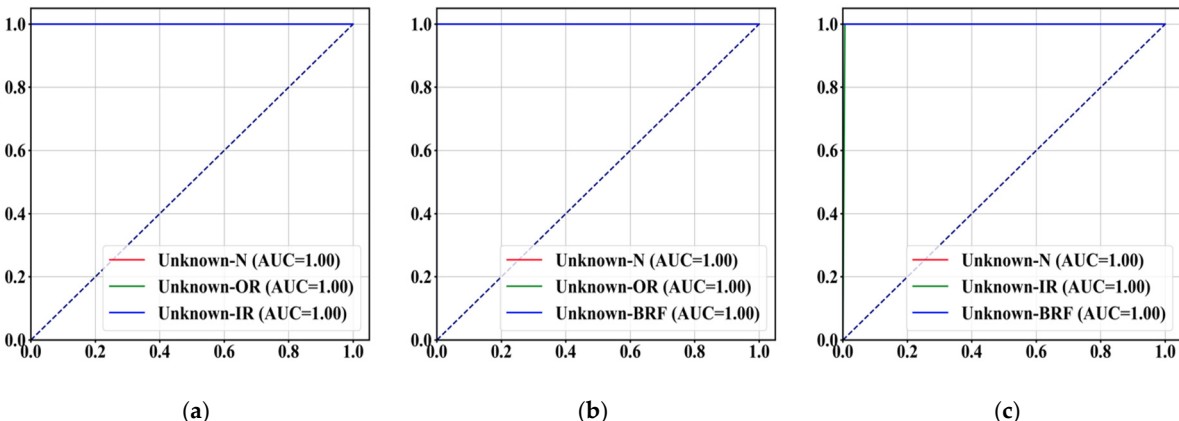

**Figure 6.** ROC/AUC of feature transferring from Ottawa to CWRU. (**a**) Hidden the labels of BRF faults. (**b**) Hidden the labels of IR faults. (**c**) Hidden the labels of OR faults.

The CWRU dataset has a length of 14,667 samples and is divided into the training set and test with a ratio of 7:3. The number of the training and test set data is 10,267 and 4400, respectively. The confusion matrix shows that though the fault categories of the target domain are different, the model can still accurately recognize the known and unknown categories. The AUC values of the three different unknown categories are all 1.00, and the AUC values of the data of the known categories are also 1.00. As CWRU has a large number of samples and has high data identification, the overall performance of CWRU as the target domain data test is better than the Ottawa data set as the target domain test data.

Tables 3 and 4 and Figure 7 show two datasets, the CWRU and Ottawa with their training time measured in hours and time reduction rate in percentage. We tried four sets of training. The first one is without transfer learning. The second one was with transfer learning, retaining the dense layer, and the classifier. The third one was with transfer learning and fine-tuning the dense layer only. The fourth and last one was with transfer learning and retaining the SoftMax plus (SP) classifier only. The graph just shows the same data in percentages. From the line chart, we use the model training time without knowledge transfer as the basis for comparison. The experimental results show that training the fully connected layer and the classifier is the most time-consuming but still faster than training from scratch. Only training the classifier is the most time-efficient, training to predict the CWRU dataset with a 34% time-saving and to predict the Ottawa dataset with a 68% time-saving.

**Table 3.** Training time measured in hours (Unit: Hour).

| Training Time (Hour) | WO/TL | W/TL (F9, F10, F11, SP) | W/TL (F9, F10, F11) | W/TL (SP) |
|---|---|---|---|---|
| Predict CWRU | 4.4 | 3.68 | 3 | 1.5 |
| Predict Ottawa | 2.55 | 2.38 | 1.88 | 1.73 |

**Table 4.** Training time reduction in percentage (Unit: %).

| Training Time (Hour) | WO/TL | W/TL (F9, F10, F11, SP) | W/TL (F9, F10, F11) | W/TL (SP) |
|---|---|---|---|---|
| Predict CWRU | 100% | 84% | 68% | 34% |
| Predict Ottawa | 100% | 93% | 74% | 68% |

The experiments have three settings. First, only the output classifier is retrained; we reuse the pre-trained model as the feature extraction mechanism. The output layer is first removed, and the entire network is used as a fixed feature extractor for the new

dataset. Second, we retrain the top densely connected classifier: using the architecture of the pre-trained model, keeping the initial weights on the convolutional base, and adding higher dense and classification layers. Perform random initialization of all weights and retrain the model on the new dataset. At this stage, data augmentation is optional. Last, fine-tuning: After training the model with the new dataset, select and freeze some layers and retrain other top layers.

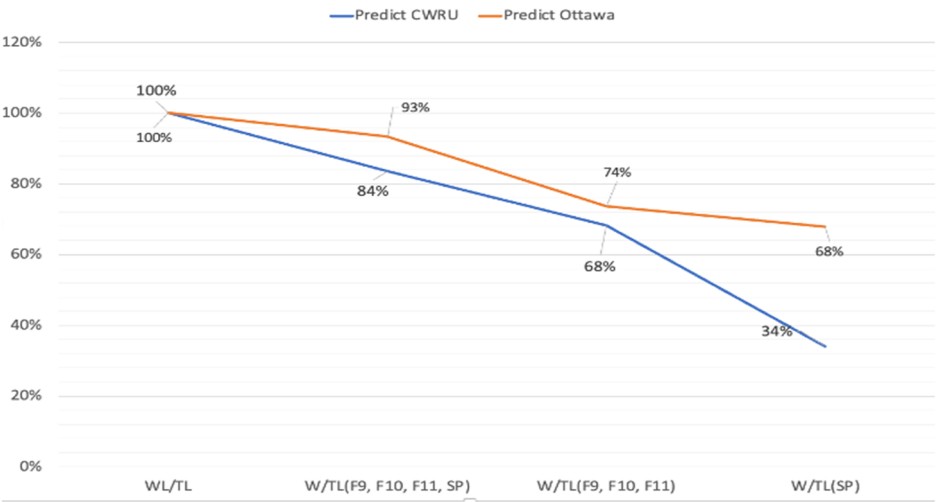

**Figure 7.** The result of training time reduction using feature transferring from Ottawa to CWRU.

Figure 8 shows the experimental results demonstrating the effectiveness and efficiency of knowledge transfer. In the bar chart, we use the accuracy of model predictions without knowledge transfer as a basis for comparison. From the experimental results, using CWRU to predict Ottawa obtains the average accuracy of 99.1% and 89.4% for known and unknown classes, respectively. The average accuracy of known and unknown classes is 99.2% and 98.2% by using Ottawa to predict CWRU, respectively. The experimental results show three points: First, the TPCCNN method inherits the advantages of the original PCCNN in distinguishing known and unknown categories with high accuracy. Second, the difference in prediction accuracy between the models with and without knowledge transfer is almost less than 1.5%. Last, using CWRU feature extraction to train and predict Ottawa's model is even more accurate than using Ottawa's model trained from scratch.

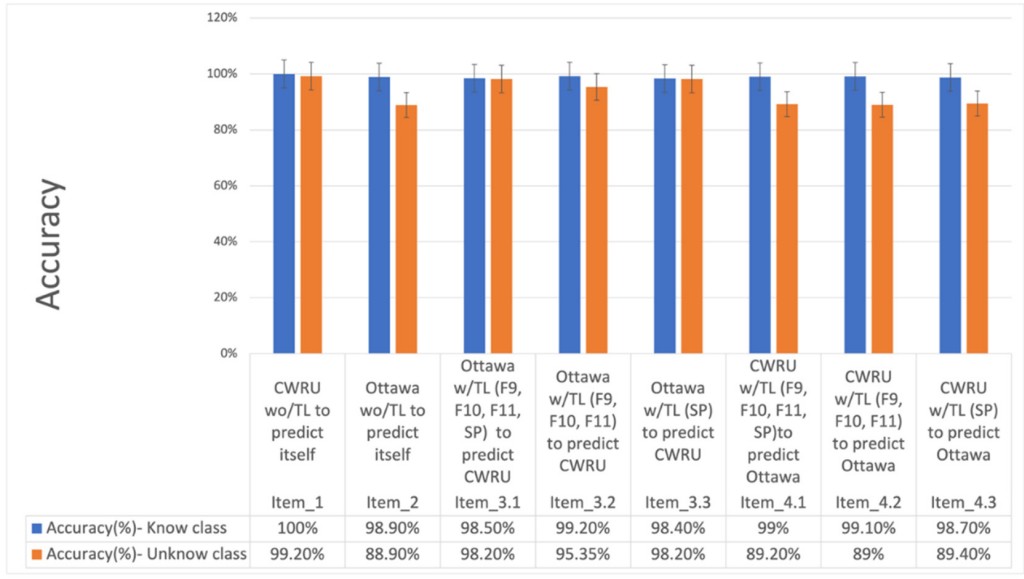

**Figure 8.** The experimental results demonstrate the effectiveness and efficiency of knowledge transfer.

## 5. Conclusions

We propose a new Transfer-learning based on Probability Confidence CNN (TPCCNN) model, which can be employed to make modeling and fault diagnosis for rotating machinery. The experimental results show the ability of the proposed approach in detecting and recognizing faults efficiently. Two public open-source datasets are used in the experiment to verify the efficiency and robustness of the TPCCNN model.

The experimental result reveals the following: First, using CWRU to predict Ottawa obtains the average accuracy of 99.1% and 89.4% for known and unknown classes, respectively. The average accuracy of known and unknown categories of Ottawa to predict CWRU is 99.2% and 98.2%, respectively. Second, the method inherits the advantages of the original PCCNN in distinguishing known and unknown categories. Third, similar feature sets can be applied to reduce the training time by 34% of CWRU and 68% of Ottawa by means of retraining and parameter fine-tuning of fully connected layers.

It is found that the proposed approach is an efficient way to detect and recognize faults. Based on the result, future work focuses on real-time fault diagnosis, strengthening the transfer learning model and making the model more adaptive.

In the future, it can be used in the fourth industrial revolution's Prognostic and Health Management (PHM) and smart buildings' operation and facilities management (FM), such as managing predictive diagnostics and maintenance of equipment like generators, pumps, etc.

**Author Contributions:** Writing—original draft, Conceptualization, Project administration, H.-M.L.; Supervision, C.-Y.L.; Software and Data curation, C.-H.W.; Writing—review and editing, M.-J.T. All authors have read and agreed to the published version of the manuscript.

**Funding:** This research was partly supported by the grant of "Talent cultivation plan for smart manufacturing-NTUST Alliance", Ministry of Education, Taiwan. (Grant No: 111DI023).

**Acknowledgments:** Taiwan Artificial Intelligence Association (TAIA), Harbor Technology Solutions Co., Ltd. (Taiwan).

**Conflicts of Interest:** The authors declare no conflict of interest.

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
