# Peer review of "A Novel Mechanical Fault Diagnosis Based on Transfer Learning with Probability Confidence Convolutional Neural Network Model"

_applsci, doi:10.3390/app12199670_

Round 1

Author Response

Point 1: Novelty of the paper should be presented carefully in the introduction part.

Response 1: The Novelty is that employment of Transfer learning method could efficiently reduce the time-consumption of PCCNN model training. Our contributions about the feature reduction methods are aggregated into two categories: data-level and algorithm-level approaches. The data-level approach consists of encoding time series using FFT to clean and produce denoising input signals which offer a more efficient CNN training. In the real world, while the spectrum with heavy noise, the FFT can efficiently clean the data and retrieve a smoothy results we expect. At the algorithm level approach, one is the PCCNN algorithm which has a self-determined and self-learned ability to distinguish between unknown and known classes. Another is a transfer learning algorithm with adaptive convolutional layer filters and classifiers to analyze the input time series signals, including noise fluctuation. The above descriptions are given in the discussion section from 283 to 291.

More descriptions are presented from line 104 to line 109 on page 3 as follows:

In previous works, fault-based modeling and data-driven methods for known fault diagnosis have been performed. At the same time, the authors of this paper made a preliminary survey of fault detection in data-driven with unknown class and transferred learning in similar datasets. This paper proposed TPCCNN (Transfer PCCNN) focuses on the vibration frequencies monitor, which can be featured through FFT and trained and transferred in the PCCNN model for further fault diagnosis for the first time.

---

Point 2: Motivation of using convolutional neural networks (CNN) between other types of neural Network should be presented clearly.

Response 2: The previous literature shows that many researchers focus on data-driven intelligent fault diagnosis (DIFD) methods. CNNs are an essential component of many existing DIFD methods. However, the commonly used DIFD methods, including traditional CNN, fail to recognize unknown classes. Therefore, this paper presents the PCCNN and improves it with a transfer learning algorithm to achieve adaptive capability. The results showed that TPCCNN can be applied to similar and cross domains. The descriptions are presented at line 75 to line 90 on page 2.

---

Point 3: Feature reduction methods can be useful in the trainning of your CNN and can be increased performance. Some comments about it should be presented in the paper.

Response 3: Thanks for valuable comments. More comments are presented from line 240 to line 263 and line 283 to line 291 as follows:

Our contributions about the feature reduction methods are aggregated into two categories: data-level and algorithm-level approaches. The data-level approach consists of encoding time series using FFT to clean and produce denoising input signals which offer a more efficient CNN training. In the real world, if the spectrum with heavy noise, the FFT can efficiently clean the data and retrieve a smoothy results we expect. At the algorithm level approach, one is the PCCNN algorithm which has a self-determined and self-learned ability to distinguish between unknown and known classes. Another is a transfer learning algorithm with adaptive convolutional layer filters and classifiers to analyze the input time series signals, including noise fluctuation.  

--

Point 4: What is free parameters in the proposed method?

Response 4: According to the proposed method TRCCNN, these adjusting parameters can be set manually from the source domain transfer to the target domain and vice versa. These are the kind of free parameters: optimizer, learning rate, momentum, split rate of original dataset, weights and bias of the unfreezing layers, etc. The total free parameters in the PCCNN model are 45. The free parameters include the split rate of original dataset setting, and the weight and bias of the unfreezing layers. Through the computing process, some parameters have some initialized value, like the dataset split rate as CWRU is 64x2 and Ottawa is 64x12, shown as line 256 to line 272 on page 7.

---

Point 5: Comparison analysis is not presented. Many papers is related to your work such as # Multi-model estimation using neural network and fault detection in unknown time continuous fractional order nonlinear systems, Transactions of the Institute of Measurement and Control, 2021, 43(3), pp. 497–509 Some remarks on comparission of your proposed merthod and this paper which is consider to fractional system (general form of the integer systems) can be useful.

Response 5:  

Thanks for valuable comments. The referred paper has been included in reference [12].

The descriptions are presented at line 72 to line 74 on page 2, and more comparison analysis descriptions are as follows:

The referred paper uses a radial basis function neural network (RBF NN) to design the observer for multi-model state estimation and then apply it to fault detection in unknown-time continuous fractional order systems. In mathematical modeling, the RBF NN uses the radial basis function as the activation function and is widely used in time series prediction, classification, system control, and function approximation.

The performance of referred RBF NN would be equivalent to the SVM with RBF kernel.

A RBF SVM would be virtually equivalent to a single-layer and double-layer RBF NN, where the first layer weights would be fixed to the feature values of all the training samples. The learning algorithm tunes only the second layer weights.

The previous work PCCNN [16] provides the performance of compared methods between the PCCNN and OCSVM (One Class SVM with RBF kernel). The results show that OCSVM performs well with accuracy in unknown categories is 99.62%, but only 41.88% in known categories. PCCNN performs well in unknown categories with 97.38% and known categories with 96.89%. Both OCSVM and PCCNN methods are unsupervised and used in abnormal detection, including novelty and outlier detection.

This paper proposed TPCCNN (Transfer PCCNN) method focuses on the vibration frequencies monitor, which can be featured through FFT and trained and transferred in the PCCNN model for further fault diagnosis. The comparison table is as follows:

Table: Comparison between referred RBF NN and proposed TPCCNN

Referred RBF NN

Proposed TPCCNN

Activation function

Gaussian or RBF

ReLU

Classifier

Sigmoid

Softmax

Fundamental

SVM

CNN-based AlexNet

Dynamic recognition 

Unknown and known classes

Unknown and known classes

Self-learning

Yes.

Yes.

Adaptive capability

No.

Knowledge Transferring

Summary:

According to the reviewer’s comment, the updated manuscript has added the material and methods and cited references. Also, an English-speaking co-worker has checked the whole article, and the authors would like to use one of the editing services listed at MDPI website later. Thank you very much for reviewing.

Reviewer 2 Report

the text in machine's picture in table is not clear. The authors should revise them.

figures 4-7 shoud be revised and each part should be added with a label.

Figure 1 is not necessary and can be removed from the MS.

What is PCCNN. I cannot find the definition in the MS.

The method for Fault Diagnosis is not clear.

Author Response

Point 1: the text in machine's picture in table is not clear. The authors should revise them.

Response 1:  

Table 2 has been revised.

Point 2: figures 4-7 shoud be revised and each part should be added with a label.

Response 2:

Figures 4-7 (3-6) have been revised . The required label has been added.

Point 3: Figure 1 is not necessary and can be removed from the MS.

Response 3:

Figure 1 has been removed.

Point 4: What is PCCNN. I cannot find the definition in the MS.

Response 4:  

PC means Probability Confidence which is employed to CNN(Convolutional Neural Network) model to improve the accuracy as reference [13].  The architecture is shown in the right side of Figure 1(Architecture of the proposed TPCCNN framework).

More descriptions are shown at line 137 to line 161 on page 4 as follows:

PCCNN is used for the computation of probabilistic confidence levels to distinguish between "known classes" and "unknown classes" of failure classes. First, being initialized with a labeled training data set, the system calculates the confidence interval and probability of each known class to evaluate the reliability probability of the statistical inference. Significance is referred to as the probability in which the estimated parameter falls within a specific range when making statistical inferences. Second, PCCNN has self-learned ability. The threshold value of each category is recorded in a vector c and defined as the probability threshold value within the normal range. when a value exceeds 1.5 times the 1st and 3rd quartile range, i.e., 1.5 x IQR, the value is classified as an outlier and placed in the unknown category.

---

Point 5: The method for Fault Diagnosis is not clear.

Response 5:  

The Fault of a rotating machine will generate saveral features such as vibration frequencies, abnormal noise etc. This paper proposed TPCCNN(Transfer PCCNN) focuses on the vibration frequencies monitor which can be featured through FFT and trained in PCCNN model for further Fault Diagnosis as shown in figure 1 .

More descriptions are shown at line 119 to line 122 on page 3 and line 164 to line 189 on page 5 and line 227 to line 280 on page 6 & 7 as follows:

2.3. TPCCNN-based Fault Diagnosis: This fault diagnostic model architecture includes data pre-processing, model pre-training, and model fine-tuning.

3.1. Dataset: The open datasets, CWRU and Ottawa, are used for model training and evaluation.

3.2. Pre-Processing: The original fault data of the source and target domain are obtained by the vibration sensor and presented as time-domain data. The fast Fourier transform (FFT) is applied to map the data into frequency domain.

3.3. Pre-Trained Model: Two models are trained and evaluated as the baseline for training and evaluation in the experiment.

3.4. Fine-tune: The experimental approach takes the architecture of a pre-trained model and then trains top layers while freezing others.

---

Summary:

According to the reviewer’s comment, the updated manuscript has added the material and methods and cited references. Also, an English-speaking co-worker has checked the whole article, and the authors would like to use one of the editing services listed at MDPI website later. Thank you very much for reviewing.

Round 2

Reviewer 1 Report

All of my comments have been done

Reviewer 2 Report

This paper can be accepted